# JAK/STAT signaling regulated intestinal regeneration defends insect pests against pore-forming toxins produced by *Bacillus thuringiensis*

Zeyu Wang[1‡], Yanchao Yang[1,2‡], Sirui Li[1], Weihua Ma[3], Kui Wang[1], Mario Soberón[4], Shuo Yan[5], Jie Shen[5], Frederic Francis[2], Alejandra Bravo[4], Jie Zhang[1]*

1 State Key Laboratory for Biology of Plant Diseases and Insect Pests, Institute of Plant Protection, Chinese Academy of Agricultural Sciences, Beijing, China, 2 Department of Functional and Evolutionary Entomology, Gembloux Agro-Bio Tech, University of Liège, Gembloux, Belgium, 3 National Key Laboratory of Crop Genetic Improvement, National Centre of Plant Gene Research, Huazhong Agricultural University, Wuhan, China, 4 Instituto de Biotecnología, Universidad Nacional Autónoma de México, Cuernavaca, Morelos, Mexico, 5 Department of Plant Biosecurity and MOA Key Laboratory of Pest Monitoring and Green Management, College of Plant Protection, China Agricultural University, Beijing, China

‡ These authors share first authorship on this work.
* zhangjie05@caas.cn

**Data Availability Statement:** The authors declare that we made all data necessary to replicate our study's findings publicly available without

## Abstract

A variety of coordinated host-cell responses are activated as defense mechanisms against pore-forming toxins (PFTs). *Bacillus thuringiensis* (Bt) is a worldwide used biopesticide whose efficacy and precise application methods limits its use to replace synthetic pesticides in agricultural settings. Here, we analyzed the intestinal defense mechanisms of two lepidopteran insect pests after intoxication with sublethal dose of Bt PFTs to find out potential functional genes. We show that larval intestinal epithelium was initially damaged by the PFTs and that larval survival was observed after intestinal epithelium regeneration. Further analyses showed that the intestinal regeneration caused by Cry9A protein is regulated through c-Jun NH (2) terminal kinase (JNK) and Janus tyrosine kinase/signal transducer and activator of transcription (JAK/STAT) signaling pathways. JAK/STAT signaling regulates intestinal regeneration through proliferation and differentiation of intestinal stem cells to defend three different Bt proteins including Cry9A, Cry1F or Vip3A in both insect pests, *Chilo suppressalis* and *Spodoptera frugiperda*. Consequently, a nano-biopesticide was designed to improve pesticidal efficacy based on the combination of *Stat* double stranded RNA (dsRNA)-nanoparticles and Bt strain. This formulation controlled insect pests with better effect suggesting its potential use to reduce the use of synthetic pesticides in agricultural settings for pest control.

restriction at the time of publication on Dryad dataset (doi:10.5061/dryad.v15dv422z).

**Funding:** This work has been funded by National Natural Science Foundation of China (32001970 to ZW). The funders had no role in study design, data collection and analysis, decision to publish, or preparation of the manuscript.

**Competing interests:** The authors have declared that no competing interests exist.

## Author summary

*Bacillus thuringiensis* (Bt) is the most successful bacteria to control insect pests through microbial pesticides and genetical modified crops. Its insecticidal proteins form pores on the midgut leading to target larvae death. However, the defense response of target insect pests is still unclear during this process. Here, we found that larvae intestine undergoes remodeling after intoxication with sublethal dose of Bt toxins. During the remodeling, the intestinal regeneration regulated by JNK and JAK/STAT signaling was induced after epithelial cells damaged by toxins. Surprisingly, this regulation is independent of EGFR pathway and *unpaired* gene that usually function in defense response in *Drosophila*. Finally, based on the findings, a mixture of dsRNA-nanoparticles and Bt bacterial strains was designed to improve pesticidal efficacy.

## Introduction

Synthetic pesticides are useful tools for protecting crops against different insect pests. However, their nonstandard use and abuse have resulted in increased pesticide residues, which remain in food, plants and soil, appearing also in processed commodities and food chain, leading to serious threats to human health and to the ecosystem balance [1–3]. In addition, several insect pests have evolved resistance to synthetic pesticides [4]. One of the most promising alternatives for the reduction of synthetic pesticides in pest control is the use of microbial bioinsecticides based on the Gram-positive bacterium *Bacillus thuringiensis* (Bt). Bt is well known for producing different insecticidal pore-forming toxins (PFTs) such as Cry and Vip toxins that attack the larval midgut cells [5]. These proteins have been widely used for the biological control of both agricultural pests and insect vectors of human diseases since the 1960s and certain toxin genes from Bt have been introduced in transgenic crops that resist insect attack [6]. Bt bacteria produce monomeric PFTs, which oligomerize upon binding to their specific receptors and assemble into transmembrane pores permeabilizing midgut cells and facilitating bacterial infection [7]. Pore formation by PFTs trigger a variety of coordinated host-cell responses as defense mechanisms [8, 9]. Double stranded RNA (dsRNA) pesticides are another safe, green and efficient emerging strategy to control insect pests or plant diseases by targeting essential insect or microbial pest genes [10, 11]. Thus, efficient formulations based on Bt and/or dsRNA technology are needed to reduce the use of synthetic pesticides in pest control. The toxicity of Bt depends on the dose and the defense responses triggered by the insect host. Thus, reducing the larval gut defense responses should enhance the toxicity of Bt PFTs.

Insect larval midgut epithelium is formed by two principal differentiated cell types: absorptive enterocytes cells (ECs, also called columnar cells) and secretory enteroendocrine cells (EE, also called goblet cells) [12–14]. When ECs are subjected to different stresses like enteric infection, they are eliminated by different mechanisms such as a regulated response of enterocyte purge [15], cell shedding [16] or apoptosis [17]. In addition, the intestinal stem cells (ISCs) play an important role in midgut homeostasis, since stress induces their cellular turnover, ensuring tissue integrity and function throughout the organism's lifetime. The replacement of damaged and aberrant cells is carried out by ISCs, whose proliferation and differentiation are tightly controlled and coordinated, preventing organ degeneration and insect death [18–20].

The Janus tyrosine kinase/signal transducer and activator of transcription signaling pathway (JAK/STAT) has emerged as a major regulator of *Drosophila* midgut regeneration [21–25]. The major substrates for JAK tyrosine phosphorylation are the STATs, which are able to bind to specific DNA sequences to activate transcription. The ligands for the JAK/STAT

pathway, such as the cytokines-Unpaired Upd2 and Upd3, are transcriptionally induced in stressed or damaged ECs and EEs, promoting division of ISCs [21, 26]. In addition, it was shown that STAT signaling also promotes and is required for EC and EE differentiation [22, 27]. However, it is not clear whether the Upd ligands are needed for this function [25]. In *Drosophila* besides the JAK/STAT signaling pathways, the ISCs are also activated in stressed epithelium through c-Jun NH (2) terminal kinase (JNK) to promote their proliferation as stem cells [27]. JNK signaling affects both ISCs and differentiated ECs. In the ECs, JNK confers stress tolerance and promotes epithelial turnover by triggering apoptosis in the damaged ECs and compensatory ISC proliferation [27, 28]. The downstream effects of JNK signaling induce Upd3 expression in ECs, which mediates JAK/STAT-dependent proliferation in adjacent ISCs [20, 29]. JNK activation in damaged ECs can also kill these cells, potentiating midgut turnover [28, 30]. JNK activation in ISCs may have different functions, such as activating stress-responsive genes, protection from oxidative damage [21], stimulating proliferation [21, 31], and altering ISC differentiation by inducing Delta expression and/or promoting ISC symmetric divisions [31–33]. Otherwise, early work on *Drosophila* revealed a central role for Delta/Notch signaling in directing the differentiation of ISC progeny [12, 34–36]. High level Notch in enteroblasts (EBs) raised by Delta from ISCs, promotes EBs differentiated to ECs; low level Notch restricts cell cycle in ISCs [29]. However, the potential participation of these pathways leading to midgut cell regeneration after Cry toxin intoxication in lepidopteran larvae, which are major pests of different crops, has not been analyzed.

In this work, we identified that the JNK/JAK/STAT is involved in regulating the midgut regeneration triggered by Bt PFTs pore formation in lepidopteran larvae. In addition, we showed that silencing JAK/STAT pathway by RNAi increased the toxicity of different PFT in two different lepidopteran species. Based on these findings, a novel and efficient formulation involving the mixture of Bt and ds*Stat* nano-pesticide was developed to control striped stem borer and fall armyworm larvae. This novel strategy for formulating Bt strains along with dsRNA that repress the insect defense response could be used for the efficient control of certain crop pests that are not highly susceptible to Bt PFTs.

## Results

### Remodeling of insect larvae midgut tissue induced by pore-forming toxins

Previously, it was reported that after exposure of *Drosophila* larvae to sublethal doses of PFT, a midgut thinness was observed, due to extrusion of the EC apical cytoplasm, into the midgut lumen, including some organelles such as mitochondria, followed by midgut tissue recovery of its normal thickness after few hours of toxin damage [15]. Here, the mortality of 2-day old and $3^{rd}$ instar larvae of the rice insect pest, striped stem borer (*Chilo suppressalis*) after ingestion of sublethal doses of Cry9Aa toxin from Bt were analyzed (Fig 1A and 1B). When a sublethal dose of Cry9Aa toxin (200 μg per g of diet) corresponding to a dose that kill 20% of the larvae (lethal concentration value = $LC_{20}$) was used to feed $3^{rd}$ instar larvae, important changes in midgut tissue morphology were observed. Analysis of laser scanning confocal microscopy (LSCM) images obtained from median sagittal views of intact midgut tissue showed that after 24 h of toxin ingestion (Fig 1C). A significant 20% reduction in midgut tissue thickness was observed (p<0.0001) that kept decreasing (up to 30% reduction, p<0.0001) till 48 h post-feeding (Fig 1D). However, after 96 h of toxin ingestion, the intestine was restored to normal levels (Fig 1C and 1D). A similar tendency, with a moderate level of tissue remodeling was observed when a 10-fold lower concentration of Cry9Aa, corresponding to a dose that kill 2% of the larvae ($LC_2$) was assayed (20 μg per g of diet) (Fig 1C and 1D), suggesting that the midgut tissue remodeling induced by the PFT was dependent on toxin concentration. As a negative control,

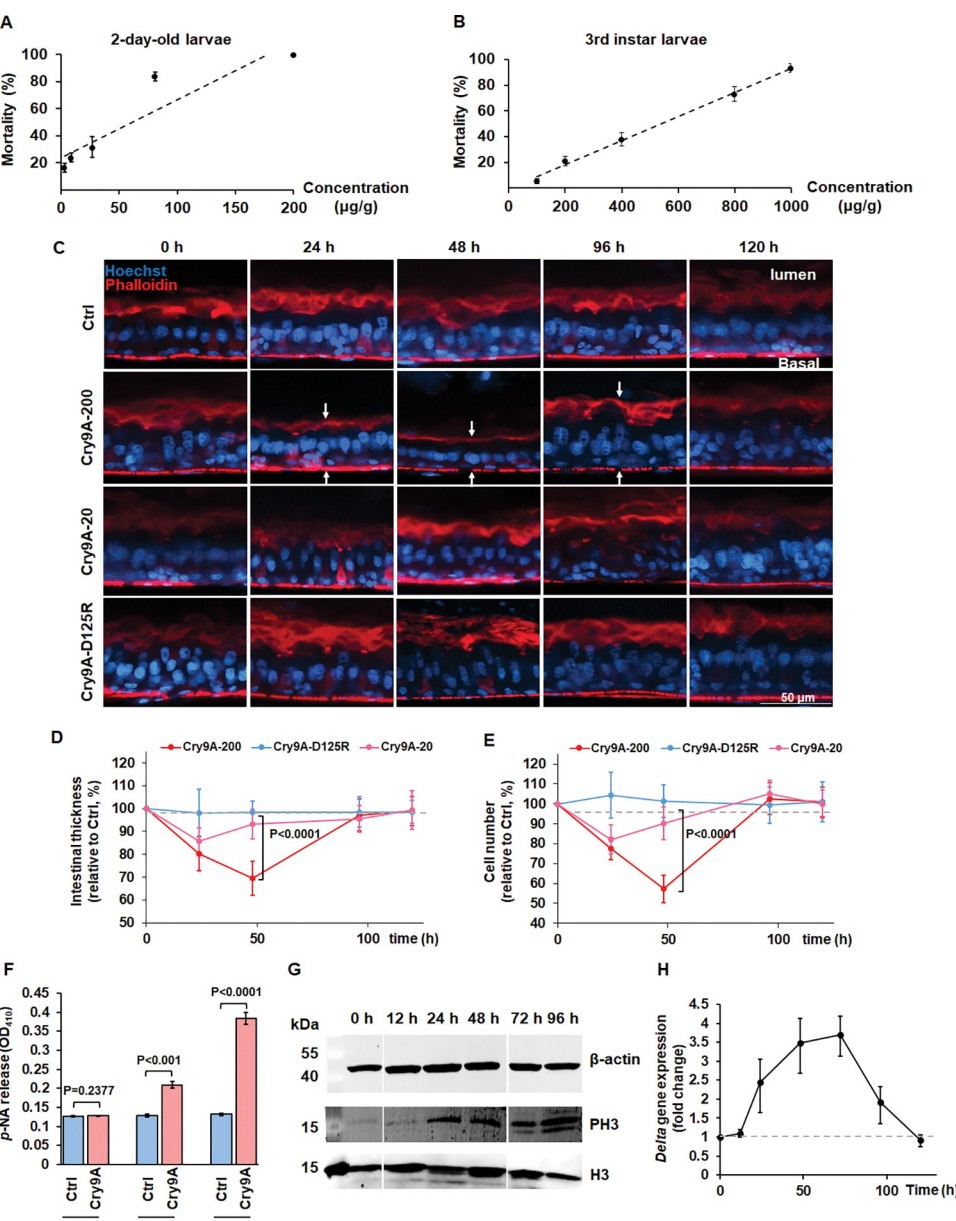

**Fig 1. Cry9Aa pore-forming toxin induced intestinal remodeling in insect larvae.** (A and B) Insecticidal toxicity assays of Cry9Aa against 2-day-old (A) or 3rd instar (B) larvae of rice striped stem borer. Thirty-five larvae were analyzed with each toxin concentration and three replicates were performed. The error bar represents ±SEM. (C) Sagittal view representative images obtained from midgut tissue of rice striped stem borer larvae treated with Cry9Aa observed in the LSCM. Phalloidin-647 (red color) stains F-actin present in the brush border (upper arrows) and in the visceral muscles (lower arrows). Hoechst33342 (blue color) labels nuclei. Control (Ctrl) samples were treated with 50 mM $Na_2CO_3$ buffer pH 10, that was the buffer used for solubilization of Cry9Aa protein. (D and E) Quantification of the fluorescent signal of intestinal thickness (D) and cell number (E) obtained after treatment with sublethal doses of Cry9Aa toxin (200 μg per g of diet) corresponding to $LC_{20}$ (Cry9Aa-200) and Cry9Aa toxin (20 μg per g of diet) corresponding to $LC_2$ (Cry9Aa-20). Toxin-free treated larvae intestines were measured as 100%. $n$ = 36 images per treatment. $P$ values were calculated by two-sided Student's t-test, $P < 0.0001$ shows statistically significant differences. The error bar represents ±SEM. (F) Analysis of p-nitroanalide release into the midgut lumen of larvae that were treated with Cry9Aa. Control (Ctrl) samples were treated with Cry9Aa-D125R mutant protein. $n$ = 15 isolated midgut tissues from 3rd instar larvae per treatment. Three biological replicate samples performed in three independent experiments were performed. $P$ values were calculated by two-sided Student's t-test, $P < 0.001$ shows statistically significant differences. $P > 0.05$ shows no significant difference. The error bar represents ±SEM. (G) Western blot detection of phospho-histone H3 (PH3) and histone H3 (H3) in midgut tissue samples from fifteen larvae after Cry9Aa ($LC_{20}$) intoxication for 0–96 h. β-actin was detected as an internal reference. (H) Quantification of *delta* gene expression

determined by RT-qPCR in intestines isolated from fifteen larvae treated with Cry9Aa (LC$_{20}$) for 0, 12, 24, 48, 72, 96 and 120 h. The data were normalized to uninfected control intestines and elongation factor-1 (*EF-1*) gene was used as reference gene on three biological replicate samples performed in three independent experiments. The error bar represents ±SEM.

a non-toxic Cry9Aa-D125R mutant (S1 Table) did not affect the intestinal epithelium of striped stem borer larvae, compared to Cry9Aa toxin (Fig 1C and 1D).

In addition, the total number of cells found in this tissue decreased significantly after 24 h of toxin ingestion, where the lowest level was observed after 48 h (p<0.0001), followed by a restoration of cell number at 96 h (Fig 1E). This effect was not observed with the non-toxic Cry9Aa-D125R mutant, while a moderate effect was recorded when a LC$_2$ concentration of Cry9Aa was used, showing a lower loss in total cell number and a faster recovery rate (Fig 1E). These data indicate that cell loss is also dependent in activity and concentration of the PFT.

To analyze further the decreasing of intestinal thickness and cell number, the enzymatic activity of GPI-anchored aminopeptidase N (APN) in the larval gut lumen was determined. The APN protein is highly abundant in the apical membrane of midgut cells and its enzymatic activity inside the midgut lumen was reported as a marker of cell shedding [37]. APN enzymatic activity indicated that cell shedding was observed after 24 h of Cry9Aa ingestion (p<0.0001) (Fig 1F). This effect was not observed with the non-toxic Cry9Aa-D125R mutant protein used as negative control (Ctrl) in these assays.

It is known that pathogens that induce midgut tissue damage, also promote ISCs proliferation [38]. Phosphorylation of histone H3 (PH3) is considered a marker for mitotic response of ISCs [27]. PH3 was detected after Cry9Aa treatment from 24 to 96 h, that is consistent with the period of time where the number of cells were recovered, and the midgut regeneration was observed (Fig 1G). ISCs specifically induce the expression of the Notch ligand, Delta. The increased level of Delta/Notch signaling from ISCs directs EBs to differentiate into ECs [25, 39]. Up-regulation of *delta* transcript was detected from 24 to 96 h after Cry9Aa treatment in midgut tissue (Fig 1H) in accordance with ISCs proliferation (Fig 1G), suggesting that cell differentiation is activated in response to cell shedding induced by Cry9Aa PFT. Overall, these data indicate that midgut was remodeled after sublethal dose of PFT treatment, showing cell loss and midgut tissue thinness. A defense response is then activated and it was observed that after 96 h of toxin ingestion, the midgut tissue recovered to normal levels through ISCs proliferation and epithelial cell differentiation allowing larval survival.

## JAK/STAT pathway regulates midgut regeneration and larvae survival counteracting the pore-forming toxin damage

The JAK/STAT pathway is a conserved signaling pathway. It was shown that in *Drosophila* JAK kinase promotes the translocation of a STAT3-like transcription factor (STAT92E) into the nucleus for regulating specific transcriptional targets such as *Socs36E* [28]. Analysis of gene expression of *Stat* and *Socs36E* genes in midgut tissue of striped stem borer larvae, showed that both genes were significantly activated at the recovery stage after 48 h of Cry9Aa treatment, but did not change after 6 h feeding with Cry9Aa (Fig 2A). RNA silencing of *Stat* and *Socs36E* (S1A Fig) by using dsRNA incorporated into star-polycation complex (SPc) nanoparticles as reported [40] increased the larvae mortality to the treatment with Cry9Aa (Figs 2B and S2). STAT inhibition with nifuroxazide, an inhibitor that represses the constitutive phosphorylation of STAT [41], also increased the mortality of larvae to Cry9Aa treatment, supporting that JAK/STAT participates in a defense response to this PFT (Fig 2C). In addition, silencing STAT by RNAi or inhibition of its activity by nifuroxazide treatment also suppressed midgut

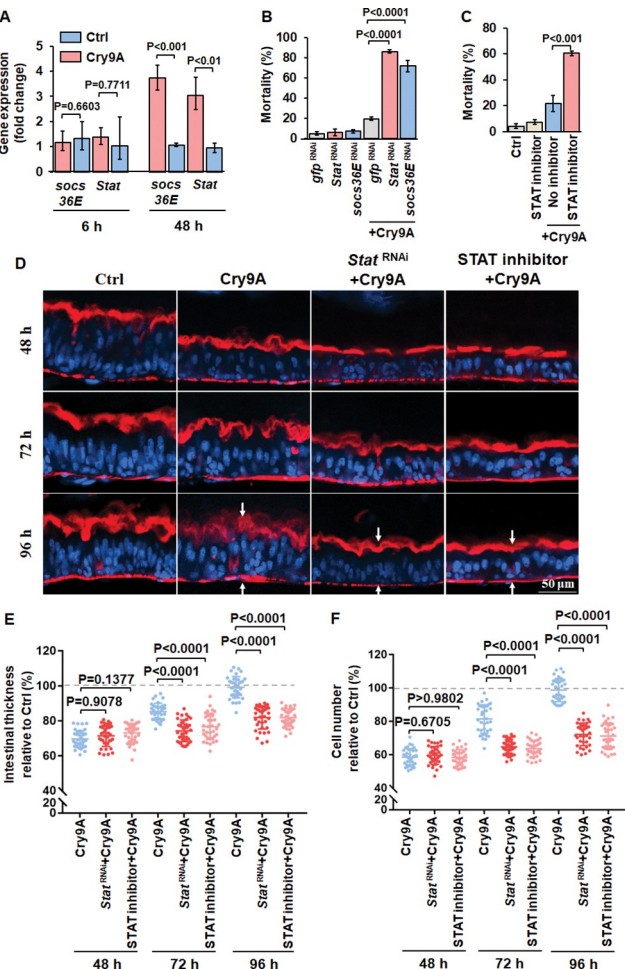

**Fig 2. JAK/STAT pathway regulates midgut regeneration and larvae survival encountering the pore-forming toxin damage.** (A) JAK/STAT pathway genes expression in the intestine from fifteen 3rd instar larvae of striped stem borer after treatment with Cry9Aa ($LC_{20}$) for different times determined by RT-qPCR. Control (Ctrl) were larvae treated with non-toxic Cry9Aa-D125R. The data were normalized to uninfected control and *EF-1* gene was used as reference gene. The experiment was done on three biological replicate samples performed in three independent experiments. *P* values were calculated by two-sided Student's t-test, $P<0.01$ shows statistically significant differences. $P>0.05$ shows no significant difference. The error bar represents ±SEM. (B) Insecticidal toxicity assay of Cry9Aa against neonate larvae of striped stem borer after silencing JAK/STAT pathway genes (*Stat* and *Socs36E* genes) by RNAi. Thirty-five neonates were analyzed in each treatment. Three replicates were performed. $P<0.0001$ shows statistically significant difference (one-way ANOVA and Tukey's test). The error bar represents ±SEM. (C) Insecticidal toxicity assay of Cry9Aa against neonates of striped stem borer after treatment with JAK/STAT pathway inhibitor Nifuroxazide. Thirty-five neonates were analyzed in each treatment. Three replicates were performed. $P<0.001$ shows statistically significant difference (two-sided Student's t-test). The error bar represents ±SEM. Control (Ctrl) were larvae treated with non-toxic Cry9Aa-D125R. (D) Sagittal view representative images of 3rd instar larvae midgut tissues isolated from alive rice striped stem borer larvae treated with Cry9Aa ($LC_{20}$) with or without *Stat* dsRNA or nifuroxazide treatment observed in the LSCM. Hoechst33342 (blue color) labels nuclei. Phalloidin-647 (red color) stains F-actin present in the brush border (upper arrows) and in the visceral muscles (lower arrows). Control (Ctrl) were larvae treated with non-toxic Cry9Aa-D125R. (E and F) Quantification of the fluorescent signal of intestinal thickness (E) and cell number (F) obtained after Cry9Aa ($LC_{20}$) with or without *Stat* dsRNA or nifuroxazide treatment. Control (Ctrl) are midguts treated with the non-toxic Cry9Aa-D125R mutant. Toxin-free treated intestine was measured as 100%. *n* = 36 images per treatment. *P* values were calculated by one-way ANOVA and Tukey's test, $P<0.0001$ shows statistically significant differences. $P>0.05$ shows no significant difference. The large bar corresponds to the mean, whereas the smaller ones represent ±SEM.

regeneration after 48 to 96 h of Cry9Aa treatment (Fig 2D). Changes in intestinal thickness and cell number were quantified in 36 midgut images, showing that midgut regeneration and ISC proliferation were both significantly suppressed (p<0.0001) (Fig 2E and 2F).

Therefore, these data indicate that JAK/STAT pathway promotes larvae survival through activating intestinal regeneration after epithelial cell loss caused by the PFT.

## JNK signaling promotes larvae survival after treatment with pore-forming toxin

It was previously shown that JNK pathway, a mitogen activated MAPK-type kinase cascade, is activated in response to cellular stress and is also involved in compensatory cell proliferation following injury of *Drosophila* intestine [20, 25, 29]. In *Drosophila*, the *puckered* (*puc*) gene encodes a Jun N-terminal kinase phosphatase which is a potent target downstream of JNK signaling [42]. We found that both *Jnk* and *puc* genes were upregulated in midgut tissue of the larvae after 6 h treatment with of Cry9Aa and the expression levels of both genes were reduced to similar levels found in the control larvae after 48 h (Fig 3A), indicating that JNK may participate in the midgut tissue regeneration at an early stage. Repression of JNK activity with the SP600125 (an anthrapyrazole inhibitor of JNK inhibitor) [43] or silencing JNK by using dsRNA incorporated into SPc nanoparticles (S1B Fig), increased the mortality of the larvae to Cry9Aa toxin treatment (Figs 3B, 3C and S2), indicating that JNK pathway also participates in the defense response to Cry9Aa action.

In relation to EGFR signaling pathway, it was reported that this pathway is essential to promote ISC proliferation under both normal and stress conditions [38, 44]. However, RNAi silencing of *Egfr* (Fig 3B) did not modify the larval mortality to Cry9Aa, although up-regulation of *Egfr* gene expression was observed at early stage (6 h) of Cry9Aa intoxication (Fig 3A and 3B).

Furthermore, silencing the JNK expression by using dsRNA incorporated into SPc nanoparticles or inhibition of JNK activity by specific anthrapyrazole inhibitor also suppressed midgut regeneration after 48 to 96 h of Cry9Aa treatment (Fig 3D–3F) without affecting the initial responses of reduction of midgut tissue thickness and total cell number. Thus, JNK signaling pathway instead of EGFR pathway promotes larvae survival through activating intestinal regeneration after epithelial cell shedding damaged by pore-forming toxin.

## JNK regulated JAK/STAT signaling induces cell proliferation and differentiation

Analysis of the temporal expression of *Jnk* gene showed that this gene was up-regulated in midgut of striped stem borer larvae after 1.5 h of Cry9Aa ingestion and reached its highest expression value after 12 h (Fig 4A). The expression of *Stat* gene also increased showing its highest expression at 48 h (Fig 4A), corresponding with the midgut recovery (Fig 2D–2F). According to the observed gene expression patterns, the relationship between STAT and JNK was further analyzed through RNAi at their corresponding time points. The expression of *Stat* in the larval midgut after 48 h treatment with Cry9Aa was suppressed after silencing *Jnk* by RNAi, while the expression of *Jnk* after 6 h treatment with Cry9Aa toxin was not affected when *Stat* gene was silenced by RNAi (Fig 4B), indicating that JNK is located upstream of STAT, regulating STAT expression.

Silencing *Jnk* or *Stat* by using dsRNA incorporated into SPc nanoparticles, also resulted in down-regulation of *delta* mRNA level (Fig 4B), indicating that *delta* expression, that is involved in the activation of epithelial cell differentiation, is regulated by JNK and STAT. In addition, immunofluorescence analysis showed that PH3 was detected using anti-Phospho-

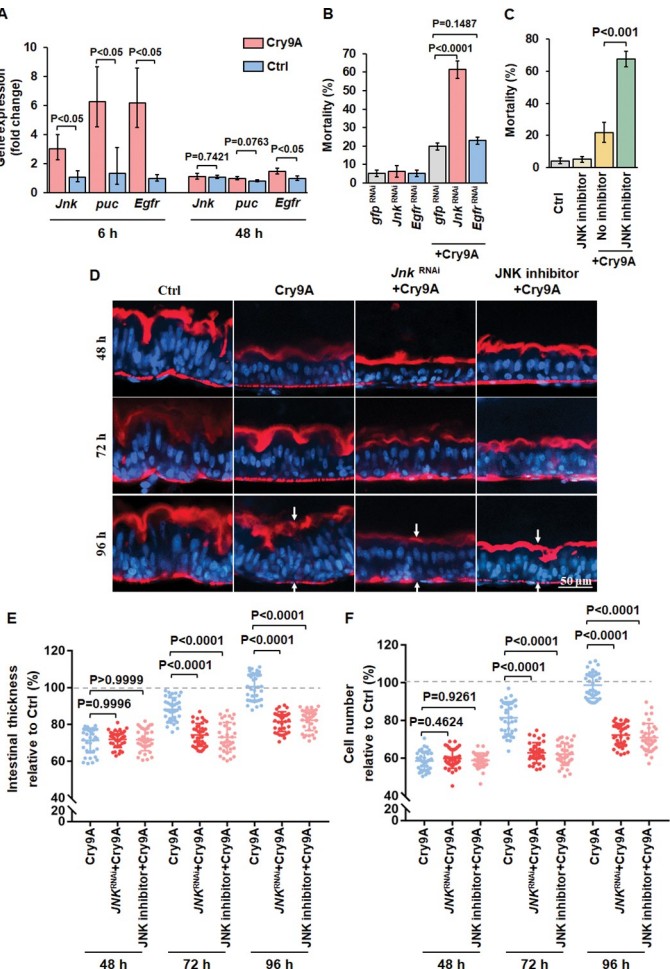

**Fig 3. JNK rather than EGFR signaling promotes larvae survival against pore-forming toxin.** (A) *Jnk*, *puc* and *Egfr* genes expression in the intestine from fifteen 3$^{rd}$ instar larvae of striped stem borer after treatment with Cry9Aa (LC$_{20}$) for different times determined by RT-qPCR. Control (Ctrl) were larvae treated with non-toxic Cry9Aa-D125R. The data were normalized to uninfected control and *EF-1* gene was used as reference gene. The data were from triplicate biological samples performed in three independent experiments. *P* values were calculated by two-sided Student's t-test, *P*<0.05 shows significant difference. *P*>0.05 shows no significant difference. The error bar represents ±SEM. (B) Insecticidal toxicity assay of Cry9Aa against neonate larvae of striped stem borer after silencing JAK/STAT pathway genes (*Jnk* and *egfr*) by RNAi. Thirty-five neonates were analyzed with each treatment. Silencing GFP was used as negative control. Three replicates were performed. *P*<0.001 shows statistically significant difference (one-way ANOVA and Tukey's test). The error bar represents ±SEM. (C) Insecticidal toxicity assay of Cry9Aa (LC$_{20}$) against neonates of striped stem borer after treatment with JNK pathway inhibitor SP600125. Thirty-five neonates were analyzed with each treatment. Three replicates were performed. *P*<0.001 shows statistically significant difference (two-sided Student's t-test). The error bar represents ±SEM. Control (Ctrl) were larvae treated with non-toxic Cry9Aa-D125R. (D) Sagittal view representative images of midgut tissues isolated from 3$^{rd}$ instar alive striped stem borer larvae treated with Cry9Aa (LC$_{20}$) with or without *Jnk* dsRNA or SP600125 inhibitor treatment observed in the LSCM. Hoechst33342 (blue color) labels nuclei. Phalloidin-647 (red color) stains F-actin present in the brush border (upper arrows) and in the visceral muscles (lower arrows). Control (Ctrl) were larvae treated with non-toxic Cry9Aa-D125R. (E and F) Quantification of the fluorescent signal of intestinal thickness (E) and cell number (F) obtained after Cry9Aa (LC$_{20}$) with or without *Jnk* dsRNA or SP600125 inhibitor treatment. Control (Ctrl) are midguts treated with the non-toxic Cry9Aa-D125R mutant. Toxin-free treated intestine was measured as 100%. *n* = 36 images per treatment. *P* values were calculated by one-way ANOVA and Tukey's test, *P*<0.0001 shows statistically significant differences. *P*>0.05 shows no significant difference. The large bar corresponds to the mean, whereas the smaller ones represent ±SEM.

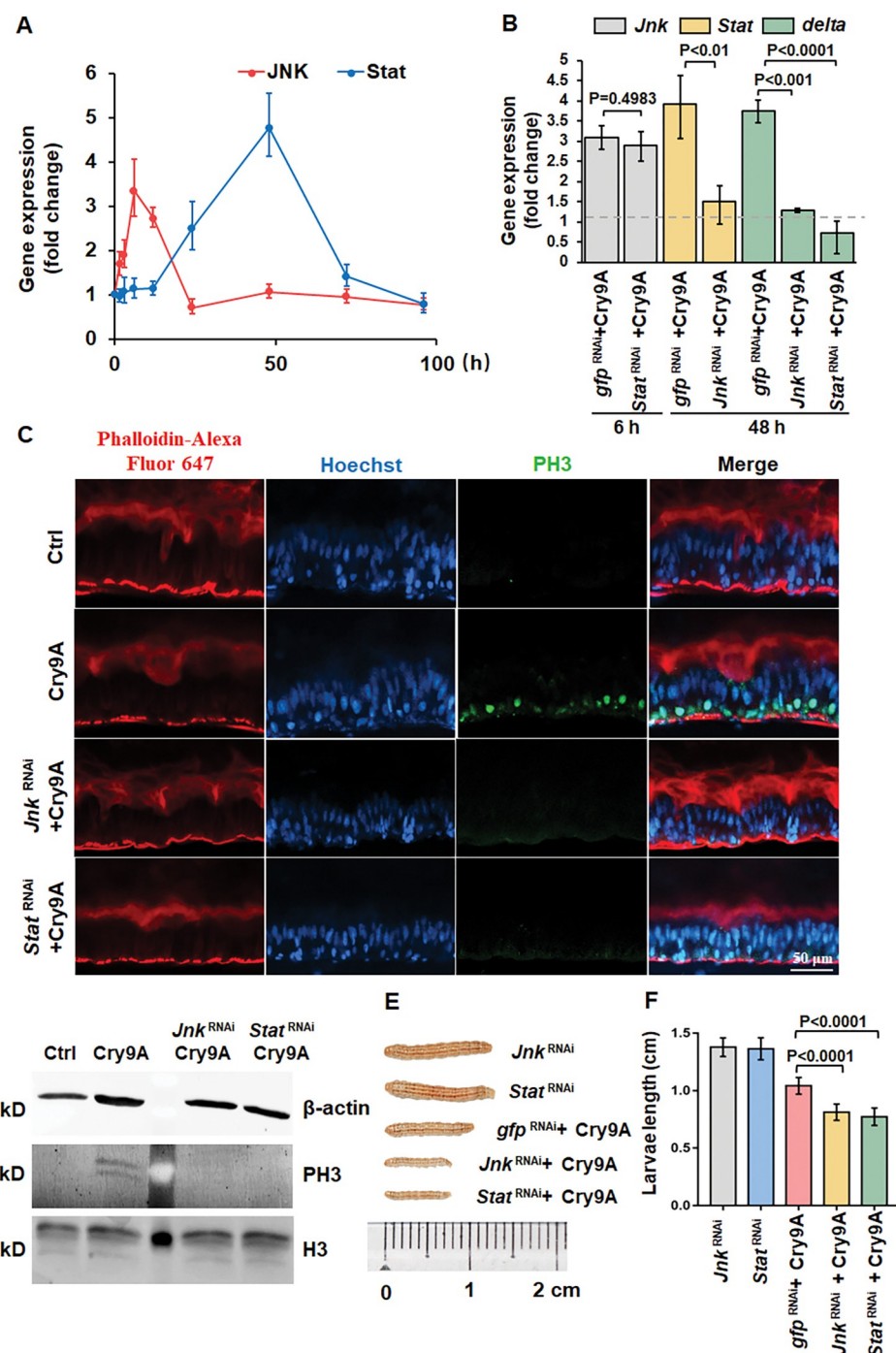

**Fig 4. JNK regulated JAK/STAT signaling induces cell proliferation and differentiation.** (A) *Jnk* and *Stat* gene expression determined by RT-qPCR in intestines isolated from 3rd instar striped stem borer larvae treated with Cry9Aa ($LC_{20}$) for different times. The data were normalized to uninfected control and *EF-1* gene was used as reference gene. Three biological replicates and three independent experiments were performed. The error bar represents ±SEM. (B) *Jnk*, *Stat* and *delta* gene expression determined by RT-qPCR in intestines isolated from silenced larvae treated with Cry9Aa ($LC_{20}$) for 6 and 48 h. Gene expression detected in toxin-free treated intestine was measured as 100%. Fifteen 3rd instar larvae were analyzed after silencing for each treatment. The data were normalized to uninfected control and *EF-1* gene was used as reference gene. Silencing GFP was used as negative control. Three biological replicate samples and three independent experiments were performed. *P*<0.05 shows statistical differences. *P*>0.05 shows no significant difference. The error bar represents ±SEM. (C) Sagittal view representative images of midgut tissues isolated from Cry9Aa ($LC_{20}$) treated 3rd instar alive striped stem borer larvae pretreated with or without *Jnk* or *Stat* dsRNA

treatment observed in LSCM. Six midgut tissues were analyzed and phospho-histone H3 (PH3) was detected by using a specific anti-PH3 antibody followed by FITC labeled secondary antibody (green color). Hoechst33342 (blue color) labels nuclei. Phalloidin-647 (red color) labels cytoskeleton. Control (Ctrl) were larvae treated with the non-toxic Cry9Aa-D125R. (D) Western blot detection of PH3 in midgut tissue samples from fifteen 3$^{rd}$ instar larvae after Cry9Aa (LC$_{20}$) intoxication after pretreatment with or without *Jnk* or *Stat* dsRNA. β-actin was detected as an internal reference. Histone H3 (H3) was detected to show that the loading quantity of H3 was similar. Control (Ctrl) were larvae treated with non-toxic Cry9Aa-D125R. (E) Representative images of seventy-two larvae treated with Cry9Aa (LC$_{20}$) for 120 h after treating with or without silencing *Jnk* or *Stat*. Negative control (Ctrl) was treated with the non-toxic Cry9Aa-D125R. (F) Quantification of length of seventy-two larvae treated with Cry9Aa (LC$_{20}$) for 120 h after treating with or without silencing *Jnk* or *Stat*. *P* values were calculated by one-way ANOVA and Tukey's test, *P*<0.0001 shows statistically significant differences.

Histone H3 antibody in ISCs after 48 h treatment with Cry9Aa toxin (Fig 4C), but the PH3 signal disappeared when *Jnk* or *Stat* were silenced by RNAi (Fig 4C). These data were confirmed by western blot assays (Fig 4D), indicating that the activation of ISC mitosis induced by Cry9Aa was regulated by both JNK and JAK/STAT signaling. We notice that although the intestinal homeostasis in the larvae was recovered after 96 h of treatment with sublethal doses Cry9Aa, the surviving larvae after 120 h grew slower than control larvae without toxin treatment (Fig 4E and 4F) suggesting that the defense process is energy consuming. However, the *Jnk* or *Stat* silenced survivals after Cry9Aa ingestion grew even smaller after 120 h due to the increased effect of toxin action.

## JAK/STAT regulated intestinal regeneration against different pore-forming toxins in another insect pest

We tested toxicity of other PFTs such as Cry1Fa [45] or Vip3Aa [46] after treatment with dsRNA of *Stat* in a different lepidopteran pest, the fall armyworm (*Spodoptera frugiperda*) larvae, a harmful corn insect pest [47]. The data showed that silencing *Stat* (S1C Fig) is effective to increase toxicity of both PFTs (Fig 5A). The intestinal regeneration after treatment with low dose of Cry1Fa was also observed in the larvae of fall armyworm (Fig 5B). Also, the larvae midgut tissue failed to recover when *Stat* was silenced by RNAi (Fig 5B and 5C). The intestinal thickness and cell number in the *Stat*-silenced larvae were both significantly lower than the non-silenced control larvae after 72 h treatment with Cry1Fa (Fig 5C and 5D). PH3 was also detected in the intestine after Cry1Fa treatment, suggesting that ISC proliferation was activated (Fig 5E) and the ISC proliferation was dampened by silencing *Stat* expression (Fig 5E). The western-blot results confirmed these data, showing that the phosphorylation of histone-H3 was depressed after *Stat* knockdown (Fig 5F). Finally, *delta* gene expression induced by Cry1Fa and Vip3Aa was also downregulated after silencing *Stat* (Fig 5G). Similar to striped stem borer, the larvae survived after 72 h of treatment with sublethal doses Cry1Fa grew faster than the larvae pretreated with *Stat* dsRNA (Fig 5H and 5I) suggesting that the JAK/STAT signaling affects global growth of the larvae through regulation of intestinal regeneration. Hence, these data indicate that JAK/STAT signaling participates inducing a broad response through intestinal regeneration after treatment with different PFTs in different lepidopteran species.

The proposed model shows that after midgut cells are damaged by sublethal dose of PFT, a defense mechanism response involving a JNK/JAK/STAT pathway was activated, promoting the midgut tissue regeneration by activating proliferation and differentiation of ISCs (Fig 6).

## Silencing *Stat* synergized pesticidal activity of Bt pesticide protecting rice seedlings from damage by striped stem borer

To test the potential use of silencing the *Stat* gene expression in combination with Bt toxins for efficient insect pest control, the *Stat*-dsRNA was incorporated into SPc nanoparticles [40] and

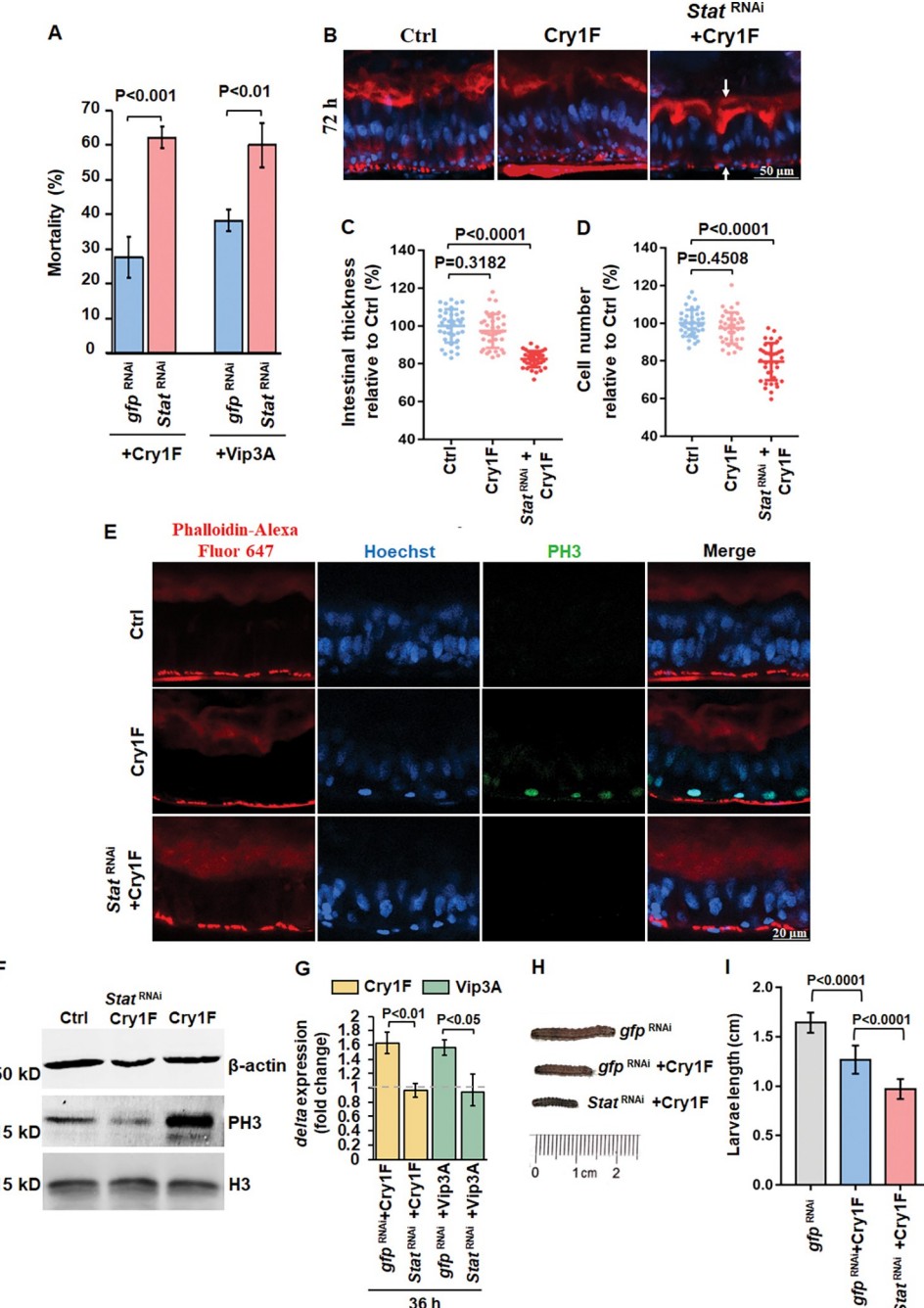

**Fig 5. JAK/STAT regulated intestinal regeneration against Cry1Fa and Vip3Aa toxins in fall armyworm larvae.**
(A) Insecticidal bioassay of Cry1Fa and Vip3Aa against neonate fall armyworm larvae that were pretreated with dsRNA of *Stat*. Thirty-five neonates were analyzed in each treatment. Three replicates were performed. *P* values were calculated by two-sided Student's t-test, *P*<0.0001 shows statistically significant differences. *P*>0.05 shows no significant difference. The error bar represents ±SEM. (B) Sagittal view representative images of 4th instar fall armyworm larvae midgut tissues isolated after 72h of Cry1Fa (LC$_{20}$) treatment with or without *Stat* dsRNA pretreatment observed in the LSCM. Hoechst33342 (blue color) labels nuclei. Phalloidin-647 (red color) stains F-actin present in the brush border (upper arrows) and in the visceral muscles (lower arrows). Control (Ctrl) were larvae treated with diet without Cry1Fa toxin. (C and D) Quantification of the fluorescent signal of intestinal thickness (C) and cell number (D) obtained after 72h of Cry1Fa (LC$_{20}$) with or without *Stat* dsRNA pretreatment. Control (Ctrl) are midguts treated with the toxin-free artificial diet. Toxin-free treated intestine was measured as 100%. *n* = 36 images per treatment. *P* values were calculated by one-way ANOVA and Tukey's test, *P*<0.0001 shows statistically significant differences. *P*>0.05 shows no significant difference. The large bar corresponds to the mean, whereas the smaller ones

represent ±SEM. (E) Sagittal view representative images of midgut tissues isolated after 36 h of Cry1Fa (LC$_{20}$) treatment from 4th instar alive fall armyworm larvae with or without *Stat* dsRNA pretreatment observed in LSCM. Six midgut tissues were analyzed and phospho-histone H3 (PH3) was detected by using a specific anti-PH3 antibody followed by FITC labeled secondary antibody (green color). Hoechst33342 (blue color) labels nuclei. Phalloidin-647 (red color) labels cytoskeleton. Control (Ctrl) were larvae treated without toxin. (F) Western blot detection of PH3 in midgut tissue samples from fifteen larvae of 4th instar fall armyworm after 36 h of Cry1Fa (LC$_{20}$) intoxication with or without *Stat* dsRNA treatment. β-actin was detected as an internal reference. Histone H3 (H3) was detected to show that the loading quantity of H3 was similar. Control (Ctrl) were larvae treated without toxin. (G) *delta* gene expression determined by RT-qPCR in intestines isolated from fifteen 4th instar fall armyworm larvae treated with Cry1Fa or Vip3Aa (LC$_{20}$) for 36 h. The data were normalized to uninfected control intestines and ecdysoneless (*ECD*) gene was used as reference gene on biological replicate samples in three independent experiments. The error bar represents ±SEM. *P* values were calculated by two-sided Student's t-test, $P<0.05$ shows statistical difference. (H) Representative images from seventy-two 4th instar larvae treated with Cry1Fa (LC$_{20}$) for 72 h with or without silencing *Stat*. Negative control (Ctrl) was treated without toxin. (I) Quantification of length from seventy-two 4th instar larvae treated with Cry1Fa (LC$_{20}$) for 72 h after treating with or without silencing *Stat*. *P* values were calculated by one-way ANOVA and Tukey's test, $P<0.0001$ shows statistically significant differences.

sprayed onto the stems and leaves of rice seedlings simultaneously with Bt serovar *fukuokaensis* strain that produces Cry9A, Cry1I, Cry9E and Vip3A proteins [48] before inoculation of striped stem borer larvae. This formulation containing both the Bt strain and SPc-ds*Stat* controlled the larvae more effectively than the control plants sprayed only with the Bt strain (Fig 7A). The toxicity of the formulation reached up to 80% mortality within three days and dead larvae were observed on the soil nearby the stems after three days of treatment (Fig 7A). Our data show that the seedlings sprayed by ds*Stat* combined with Bt strain grew much healthier and stronger than the other treatments, showing bigger leaves (Fig 7B), fewer wormholes on the stems (Fig 7C) and higher plant height (Fig 7B and 7D). In contrast, the seedlings without treatment or treated only with the dsRNA were not protected (Fig 7B–7D). The control where ds*gfp* was used in a mixture with Bt strain, showed some protection due to Bt toxins

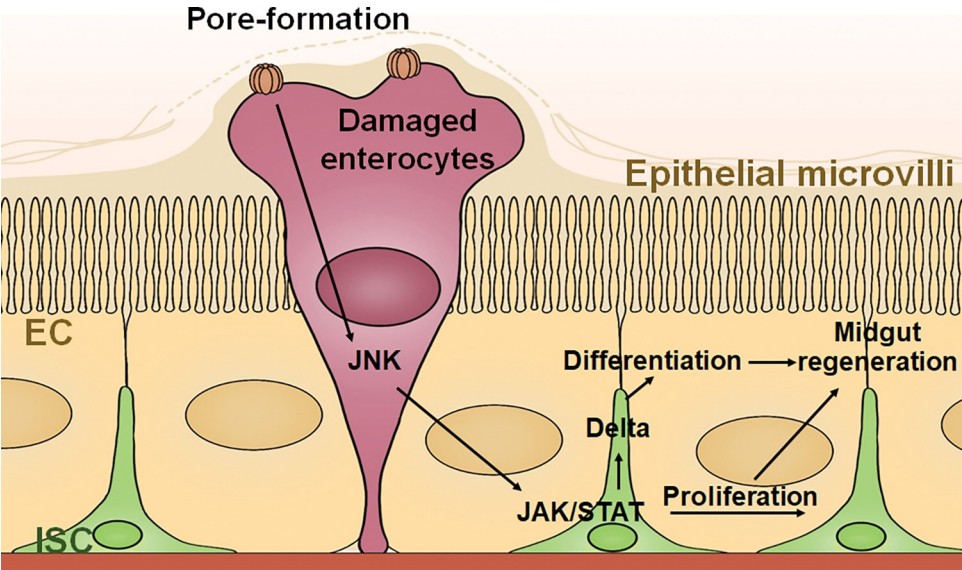

**Fig 6. Schematic model of the proposed defense mechanism activated in the larval midgut cells after intoxication with sublethal dose of PFTs.** Pore-forming toxins such as Cry9Aa, Cry1Fa and Vip3Aa produced by *Bacillus thuringiensis* (Bt) form pores on the apical membrane of larval midgut cells. Pore-formation activates JNK signaling in a short time after sublethal intoxication in enterocytes (EC). As followed, JAK/STAT signaling instead of EGFR is induced for both intestinal stem cells (ISC) proliferation and Delta regulated differentiation resulting in intestinal regeneration and larval survival.

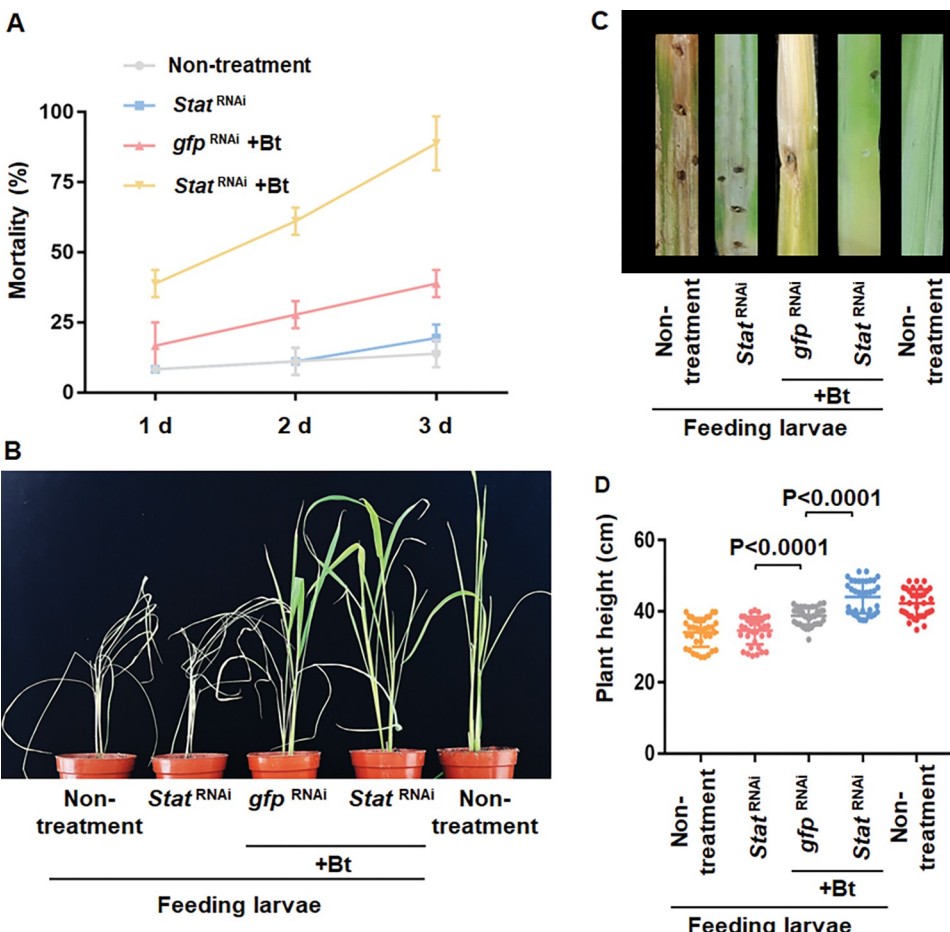

**Fig 7. Spraying *Stat* dsRNA-nanoparticles synergized Bt pesticides to protect rice seedlings from striped stem borer.** (A) Mortality of 2[nd] instar larvae of striped stem borer (six larvae per plant) on rice seedlings sprayed with the mixture of Bt *fukuokaensis* strain and *Stat*-dsRNA or *gfp*-dsRNA combined nanoparticles. Representative larvae treated with Bt strain for 3 days with or without silencing *Stat*. Negative control were plants sprayed without Bt strain. The error bar represents ±SEM. (B) Growth of rice seedlings sprayed with or without the mixture of Bt pesticide and *Stat*-dsRNA combined nanoparticles. Negative control were plants sprayed with Bt strain and *GFP*-dsRNA combined nanoparticles. Two rice seedlings at four-leaf stage per pot and two pots per treatment, with three replicates were performed. (C) Representative images of wormholes produced by the larvae that damage the rice stems, especially near the ground. The different treatments were indicated in the Fig. (D) Quantification of plant height of rice seedlings sprayed with or without mixture of Bt strain and *Stat*-dsRNA combined nanoparticles and the infected with striped stem borer larvae. *P* values were calculated by one-way ANOVA and Tukey's test. $P < 0.0001$ indicates statically significant difference. The large bar corresponds to the mean, whereas the smaller ones represent ±SEM.

action since fewer dried and frizzled leaves (Fig 7B), and fewer wormholes on the stems were found (Fig 7C) when compared with controls without Bt strain treatment.

The dsRNA designed for silencing *Stat* showed no homologous genes in the genome of *Homo sapiens* assemblies (GCF_000001405.40; GCF_009914755.1) chromosomes. *Stat* gene is relatively conserved among different lepidopteran insects, for instance the gene identity is 50.1% aligned in silkworm. However, the dsRNA we used here was designed from the specific region of *Stat* in striped stem borer showed no homology with the genes in silkworm (GCA_026075555.1) indicating its potential safety for non-target organisms. We concluded that spraying of ds*Stat* combined with Bt pesticides could be an effective and safe strategy to improve control of insect pests.

## Discussion

The PFTs produced by Bt are recognized as key virulence determinants for infecting and killing their insect hosts. In general, PFTs disrupt the plasma membrane by generating protein pores that induce uncontrolled ion exchanges between the extracellular and intracellular milieus, disturbing dramatically the cellular homeostasis [49]. Important advances in the characterization of conserved cellular repair mechanisms evolved in eukaryotic cells have been reported, allowing cells to recover from the mechanical disruption produced by PFT in their plasma membrane [9]. These defense mechanisms include clogging the pore [50], quarantining membrane damage by blebbing [51], shedding of large cytoplasm extrusions and thinning of damaged epithelia cells [52, 53], endocytosis/ exocytosis and degradation after fusion with lysosomes [8, 54]. In *Drosophila*, it was described that cell turnover rates in the midgut tissue are in constant flux as a response to different stress conditions, such as intoxication with PFT produced by infectious enteric microbes [25].

In order to identify target genes involved in midgut cell repair and turnover to set strategies for enhancing the toxicity of Bt insecticidal proteins in lepidopteran insects we characterized signal transduction pathways involved in larval midgut cell regeneration after exposure to sublethal dose of Bt Cry9Aa PFT. Larval exposed to Cry9Aa experienced midgut damage and cell loss followed by ISCs proliferation and tissue regeneration (Fig 1D–1F). These data agree with previous publications showing damage of lepidopteran midgut tissue after ingestion of Cry1A toxins [55].

We demonstrated that JAK-STAT signaling regulates intestinal regeneration in lepidopteran larvae after treatment with PFTs (Fig 2), which agrees with previous publications showing that JAK-STAT signaling pathway is a major regulator of *Drosophila* midgut regeneration [23–25]. It was shown before that when midgut tissue of *Drosophila* was stressed or damaged, ECs stimulates regeneration of ISC by triggering JAK/STAT signaling response in which Upd2 and Upd3 are transcriptionally induced in ECs and EEs, triggering STAT signaling in ISCs, promoting their division [21, 26]. Surprisingly, no significant similarity to *Upd* genes were found in the genome of the striped stem borer. One possibility is that there are other molecules delivering the signals from ECs to ISCs in striped stem borer. Another possibility is that the detached ECs was sensed by ISCs through the loss of E-cadherin as a kind of cell-cell communication [56].

Here, we showed that JNK was upregulated at early stage after infection (Fig 4A) and silencing *Jnk* by RNAi inhibited the STAT response of striped stem borer larvae (Fig 4B), suggesting that JNK signaling may initiate STAT response during intestinal regeneration. These data correlate with previous works showing that upon infection by pathogens that directly damage the gut epithelium, such as *Pseudomonas*, JNK signaling is activated in ECs [27, 30]. Our data cannot distinguish if STAT activation by JNK is direct or indirect, more studies are required to solve this question. Also, the localization of JNK and STAT signaling and identification of responsible cells remains to be determined. This information will be useful to understand the participation of this pathway in the defense response to Cry9Aa toxin.

Likewise, it was previously shown that the EGFR/Ras/MAPK pathway is required to induce EE/EC growth in *Drosophila* after midgut damage caused by the PFT haemolysin produced by *Pseudomonas entomophila* [38]. However, we found that RNAi for *Egfr* did not change the mortality of striped stem borer larvae to Cry9Aa toxin, although *Egfr* was upregulated in early stage of PFT intoxication (Fig 3A). We hypothesized that in the case of larval midgut tissue exposed to Bt Cry9Aa PFT, EGFR may be functioning independently from JNK/JAK/STAT pathway. Also, we cannot discard that EGFR is involved also in the intestinal regeneration in lepidopteran insects but other pathways compensate the silencing of *Egfr* explaining the lack of phenotype in the silenced larvae. This remains to be studied in the future.

Overall, our data shows that JNK and JAK/STAT pathways are involved in midgut tissue regeneration after damage by PFT. Also, that JAK/STAT is a conserved pathway to regulate defense in two different insect lepidopteran pests, striped stem borer and fall armyworm, to three different Bt PFTs (Cry9Aa, Cry1Fa) (Fig 5). As mentioned previously, the main objective of this work was to identify target genes to knockdown the intestine regeneration response in order to increase the toxicity of Bt Cry insecticidal proteins. Our data shows that silencing *Stat* gene inhibits the intestine regeneration after exposure to Cry insecticidal proteins enhancing the toxicity of Cry proteins to at least two different insect pests (Fig 7). Furthermore, we demonstrated that spraying this novel formulation into plants helps to prevent the damage of the insect pest. Thus, this is a novel strategy that may be used for enhancing Cry toxicity to different insect pests. Furthermore, the simultaneous expression of *Stat* dsRNA and Cry toxins in transgenic plants or in a topic formulation could provide means for an efficient control of pests that show low susceptibility to Cry toxins or to insects that have evolved resistance to these insecticidal proteins.

## Materials and methods

### Insect populations

Striped stem borer (*Chilo suppressalis*) obtained from Hubei province in China was supplied by Shennong Inc. (Hubei, China). Fall armyworm (*Spodoptera frugiperda*) obtained from Yunnan province in China was supplied by Prof. Wenjing Xu, Institute of Plant Protection, Jilin Academy of Agricultural Sciences.

### Insecticidal bioassays

The Cry9Aa3, the non-toxic Cry9Aa3-D125R mutant and the Vip3Aa11 proteins were expressed in *Escherichia coli* and purified by using Ni-chelating affinity chromatography as previously described [57]. The Cry1Fa3 protein was expressed in Bt and purified according to crystal lysis method described previously [58].

Bioassays to determine median lethal concentration values were performed on 2-day-old larvae, while additional mortality bioassays were performed on neonate larvae after two-day feeding with dsRNA by using diet incorporation assay according to previous description [57, 59]. For insecticidal mortality tests against striped stem borer, five gradient concentrations of Cry9Aa or its mutant protein in one mL were mixed thoroughly with 3-g artificial diet (supplied by Prof. Lanzhi Han and Prof. Yunhe Li from Institute of Plant Protection, Chinese Academy of Agricultural Sciences) [60] and transferred into flat glass tubes. Thirty-five larvae were placed inside each tube and three replicates were performed for each treatment. The mortality was calculated after 7 days. Lethal concentration of 20% larvae mortality ($LC_{20}$) and lethal concentration 2% larvae mortality ($LC_2$) was calculated by Probit analysis [61].

For insecticidal mortality assays against fall armyworm, a modified artificial diet based on soybean powders and wheat bran was prepared according to previous publication [62]. In the bioassays, 15 g artificial diet were incorporated with 7.5 µg Cry1Fa (0.5 µg/g) or 15 µg Vip3Aa (1 µg/g) and evenly packed into 24-well culture plates. One 2-day-old larva was placed in each well. Three replicates were performed for each treatment. The percentage of mortality was calculated after 7 days.

### Intestinal thickness and cell number determination

The midgut tissues from third instar larvae that survived were dissected after feeding with 200 µg Cry9Aa protein per g of diet for different times (1.5–120 h) and fixed overnight in 4%

paraformaldehyde at 4˚C. These midgut tissues were carefully washed three times with PBS and labeled with 1:1000 Phalloidin-iFluor 647 (Abcam, Cambridge, UK) for 1 h at room temperature (RT). After washed twice with PBS, the tissues were labeled with 10 µg/ml Hoechst33342 (Solarbio Life Sciences, Beijing, China) final concentration in PBS for 5 min. The tissues were sealed with Prolong Diamond Antifade (Invitrogen, Oregon, USA) on microscopic slides after washing three times with PBS. Images were obtained using 20X objective in the confocal LSM 980 (Zeiss, Germany). Six guts were used for each treatment and six images were taken from each midgut. The intestinal thickness measurements were based on the data of the cytoskeleton stained with Phalloidin-iFluor 647 by using ImageJ software in a total of 36 images. Midgut cell numbers were counted based on number of nuclei stained with Hoechst33342 and then dividing by the gut length as the number of nuclei per µm area. Three replicates were performed for each treatment.

## Analysis of APN release

Third instar larvae were treated with 200 µg/g Cry9Aa for 12 and 24 h. Fifteen intact midguts were dissected from alive larvae at the indicated times and transferred into 100 µl of 50 mmol/L Tris-HCl, pH 7.5. Vortexed 30 sec, centrifuged at 21,000 $xg$ for 5 min at 4˚C, and the supernatants containing contents in midgut lumen of these samples were used for APN activity analysis. Protein concentration in the samples was measured by using Bradford kit (Solarbio Life Sciences, Beijing, China). APN activity was assayed using 4 mM L-leucyl-$p$-nitroanilide as substrate in 50 mM Tris-HCl (pH 7.5) buffer. The released $p$-nitroanilide by the hydrolysis of LpNA was monitored at 405 nm for 10 min using FlexStation 3 Multi-Mode Microplate Reader (Molecular Devices, USA). The higher APN enzymatic activity was the result of an increase in intestinal cell shedding that represented intestinal damage condition. In these experiments, we used as negative control (Ctrl) a non-toxic Cry9Aa-D125R mutant protein which the mutation site is located in domain I helix α3. This mutant lost toxicity but retains the same ability to bind to the BBMV as the Cry9Aa. Control and all the treatments were performed in three repetitions.

## RNA interference assays

The DNA templates for dsRNA were cloned from cDNA of larvae midguts (information about primers and the selected gene fragments of dsRNA are described in S2 Table). The DNA was ligated into pEasy Blunt Zero vector (TransGen Biotech, Beijing, China). A 428-bp fragment of enhanced green fluorescent protein gene (*egfp*) was separately constructed [63]. The different dsRNAs were synthesized from plasmid DNA using the T7 RiboMAX Express RNAi System (Promega, Madison, USA) according to the manufacturer's instructions. The concentration of synthetic dsRNA was determined by using NanoDrop (ThermoFisher Scientific, Oregon, USA). Mixed 75 µg dsRNA with Star Polycation (SPc) complex [40] at 1:1 mass ratio in 1 mL nuclease-free water. After incubated for 15 min at RT, fifty striped stem borer or fall armyworm neonates were treated with 3 g diet mixed with the SPc-dsRNA formulation for 48 h. SPc-ds*egfp* was used as negative control. Fifteen larvae were collected for reverse transcript quantitative PCR (qRT-PCR) and the rest of the treated larvae were transferred to artificial diet containing 20 µg of Cry9Aa per g of diet for striped stem borer, or 0.5 µg Cry1Fa per g of diet or 1.0 µg of Vip3Aa /g diet for fall armyworm insecticidal bioassay. 200 µg /g Cry9Aa ($LC_{20}$) for intestinal regeneration analyses of third instar larvae of striped stem borer, or with 30 µg /g Cry1Fa or 77 µg /g Vip3A ($LC_{20}$) for forth instar larvae of fall armyworm as described above. Three repetitions were performed for each treatment.

## Reverse transcript quantitative PCR (RT-qPCR)

RNAi efficiency and gene expression were evaluated by RT-qPCR using primers described in S3 Table. Thirty 2-day-old larvae or fifteen midguts from third-instar larvae were collected together and mixed in a single sample for analysis of RNAi efficiency and for gene expression determination for each treatment. Total RNA was extracted from the samples by using Total RNA Kit I (OMEGA Bio-tek, GA, USA). Five hundred ng RNA was used for reverse transcription by using HiScript III RT SuperMix (Vazyme, Nanjing, China). Twenty-fold dilution of cDNA was used as template for qPCR by using SYBR qPCR Master Mix (Vazyme, Nanjing, China) through QuantStudio 6 (Applied Biosystems, MA, USA). Relative gene expression was calculated in relation to a reference gene, *elongation factor-1* (EF-1) for striped stem borer [64] and *ecdysoneless* (ECD) for fall armyworm larvae [65], using the $2^{-\Delta\Delta Ct}$ method [66]. Three replicates and three independent experiments were performed.

## Immunofluorescence microscopy

Six midgut tissues for each treatment were dissected from the larvae and fixed in 4% paraformaldehyde overnight at 4˚C. After washed three times with PBS, they were carefully permeabilized with 2% Triton X-100 in PBS for 2 h at room temperature. Primary antibodies were added and incubated for 2 h at room temperature, their concentrations were as follows: anti-Phospho-Histone H3 antibody 1:50 dilution (Abclonal Technology, China). After washing with PBS for three times and 5 min for each time, a secondary goat anti-rabbit-FITC antibody (1:100 dilution) (Solarbio Life Sciences, Beigjing, China) was added and incubated at RT for 2 h [67]. After washing with PBS for three times and 5 min for each time, the midgut tissues were labeled with phalloidin-iFluor 647 and Hoechst33342 as describe above. Images were obtained using confocal LSM 980 microscopes. Measurements of fluorescence intensity were done by using Image J software.

## Western blot

Total protein of fifteen midgut tissues obtained from larvae was extracted by using Protein Kit (OMEGA Bio-tek, GA, USA). The amount of total protein from midgut samples was quantified using a BCA kit (Solarbio Life Sciences, Beigjing, China). Then 35 µg protein for each treatment were resolved on SDS-PAGE gels and electrotransferred to PVDF filters. After blocking in 1% non-protein blocking solution (Sangon Biotech, Shanghai, China) for 40 min at RT, the PVDF filters were incubated for 4 h with one of the following primary antibodies anti-β-actin (ABclonal Technology, Wuhan, China) at 1: 30000 dilution, anti-Histone H3 (ABclonal Technology, Wuhan, China) at 1: 3000 dilution and anti-Phospho-Histone H3 (ABclonal Technology, Wuhan, China) at 1:1000 dilution in PBS containing 0.1% Tween-20 (PBST). After washing with PBST three times for 10 min each, the PVDF filters were incubated with corresponding secondary Goat anti-rabbit antibody labelled with FITC (Solarbio Life Sciences, Beigjing, China) in PBST at 1:1000 dilution for 1 h at RT. Images were obtained with a Typhoon 9410 scanner (GE Healthcare, USA) and analyzed with Image J software.

## Pot experiment

Two rice seedlings (*Oryza sativa* L.) at four-leaf stage per pot were utilized for spraying assay. Five mL formulation containing $1.8 \times 10^9$ cfu Bt serovar *fukuokaensis* strain with *cry9Aa* gene preserved in our laboratory [48] mixed with 60 µg dsRNA samples embedded in SPc at 1:1 mass ratio as previously reported [40] were sprayed on the seedlings. After 1 h, six second-instar striped stem borer larvae were placed on each plant. Two pots were used for each

treatment. Three replicates were performed. We analyzed wormholes and dead insects daily, and calculated the plant heights on the seventh day.

## Statistical analysis

The data required to replicate all study findings reported in the article is published and publicly available on Dryad dataset [68]. All the experiments were repeated at least three times and the statistical analyses were performed using Graphpad Prism 7.0 software (Graphpad, San Diego, CA). Pairwise comparisons were performed by Student's t-test. One-way analysis of variance (ANOVA) was used in comparing the multiple groups, and Tukey's test was used as a post-hoc test (percentages via log transformation).

## Supporting information

**S1 Fig. RNAi efficiency determined by RT-qPCR in neonate larvae treated for 48 h with the corresponding dsRNA.**
(TIF)

**S2 Fig. The full dose-response curves of Cry9A protein against *Chilo suppressalis*.**
(TIF)

**S1 Table. Median lethal concentration of Cry9A and its mutant proteins.**
(XLSX)

**S2 Table. Primers of synthesized dsRNA.**
(XLSX)

**S3 Table. Primers of RT-qPCR.**
(XLSX)

## Acknowledgments

Prof. Lanzhi Han and Prof. Yunhe Li from Institute of Plant Protection, CAAS for supplying artificial diet for striped stem borer; Prof. Wenjing Xu for supplying larvae of fall armyworm.

## Author Contributions

**Conceptualization:** Zeyu Wang, Jie Zhang.

**Data curation:** Jie Zhang.

**Formal analysis:** Zeyu Wang, Yanchao Yang, Kui Wang.

**Funding acquisition:** Zeyu Wang, Jie Zhang.

**Investigation:** Zeyu Wang, Yanchao Yang, Sirui Li.

**Methodology:** Zeyu Wang, Weihua Ma, Kui Wang.

**Project administration:** Zeyu Wang, Jie Zhang.

**Resources:** Weihua Ma, Shuo Yan, Jie Shen, Jie Zhang.

**Software:** Yanchao Yang, Kui Wang.

**Supervision:** Frederic Francis, Jie Zhang.

**Validation:** Yanchao Yang, Sirui Li.

**Visualization:** Zeyu Wang, Yanchao Yang.

**Writing – original draft:** Zeyu Wang, Yanchao Yang.

**Writing – review & editing:** Mario Soberón, Alejandra Bravo.

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
