## [Decision Letter · Decision Letter 0]

15 Aug 2023

Dear Professor Zhang,

Thank you very much for submitting your manuscript "JAK/STAT signaling regulated intestinal regeneration defends insect pests against pore-forming toxins produced by Bacillus thuringiensis" for consideration at PLOS Pathogens. As with all papers reviewed by the journal, your manuscript was reviewed by members of the editorial board and by several independent reviewers. In light of the reviews (below this email), we would like to invite the resubmission of a significantly-revised version that takes into account the reviewers' comments.

Three reviewers and myself have looked over the manuscript. All four of us see positive aspects to the manuscript. One reviewer had much more significant issues than the other two and I believe that reviewer has valid points. I am therefore sending this as Major Revision. Please address all comments from all reviewers. THank you.

We cannot make any decision about publication until we have seen the revised manuscript and your response to the reviewers' comments. Your revised manuscript is also likely to be sent to reviewers for further evaluation.

Sincerely,

Raffi V. Aroian

Academic Editor

PLOS Pathogens

Michael Wessels

Section Editor

PLOS Pathogens

Kasturi Haldar

Editor-in-Chief

PLOS Pathogens

orcid.org/0000-0001-5065-158X

Michael Malim

Editor-in-Chief

PLOS Pathogens

orcid.org/0000-0002-7699-2064

Three reviewers and myself have looked over the manuscript. All four of us see positive aspects to the manuscript. One reviewer had much more significant issues than the other two and I believe that reviewer has valid points. I am therefore sending this as Major Revision. Please address all comments from all reviewers. THank you.

Reviewer's Responses to Questions

**Part I - Summary**

Reviewer #1: The manuscript proposes the reduction of gut defences to Bt toxins will enhance their effects. The involvement of JNK/JAK/STAT in midgut regeneration is established in Chilo suppressalis and JAK/STAT and JNK silencing is shown to potentiate toxin action through a combination of silencing and inhibition approaches. No effect was seen on Cry9A susceptibility with Egfr silencing. A combined Bt/RNAi XXX was developed, tested and shown to be more effective than Bt treatment alone.

This is a very thorough and well carried out study. I have only very minor suggestions for the improvement of the work (see below). There are a few minor errors in English grammar/phrasing but none that obscure meaning.

Reviewer #2: In this manuscript, the authors explore the mechanisms underlying insect resistance to Cry toxins, using striped borers and armyworms. At low toxic doses, they find an increase in intestinal thinning and toxicity when insect JNK or STAT92E is blocked by inhibitors or dsRNA. They treat rice plants with a mix of dsRNA targeting JNK and STAT92E in combination with Bt, and find that the combination has increased efficacy compared to Bt alone. Based on these results, the authors propose a mechanism by which JNK activation in dying epithelial cells triggers Stat activation in stem cells, leading to a regenerative response.

The strengths of the manuscript include the importance of Cry toxins, and the need to identify and target survival mechanisms in insects. The authors propose a pathway and show proof of concept in a treatment modality.

The weaknesses of the manuscript include the extreme doses used in the treatment, incorrect statistical analysis, missing controls, contradictory LC50 data, and a failure to rigorously test the proposed mechanism by determining if these signaling proteins act in parallel, additively, synergistically, and where they function.

Reviewer #3: This is an interesting study that investigates the role of healing of the midgut epithelium in pest insect tolerance to Cry toxins from Bacillus thuringiensis. It is heavily motivated by studies in Drosophila and although some conservation in stem cell induction pathways might be conserved across Diptera and Lepidoptera, it is not clear whether there were other target genes from Drosophila that were investigated and not found to have the same effect in the lepidopteran species. Yet the experimental findings from the lepidopteran genes appear to be significant.

**Part II – Major Issues: Key Experiments Required for Acceptance**

Reviewer #1: None

Reviewer #2: Key experiments needed

The authors need to test their proposed model in Fig 6, especially the diverse localization of necessary JNK and STAT92E signaling, and the cell types responsible.

The proposed work does not make it clear if these are additive effects, synergistic effects, or if these are parallel pathways. While Fig 4 suggests JNK may be necessary for Stat signaling, it could be that loss of JNK has indirect effects. For example, if the mode of cell death changes, perhaps key molecules are not secreted to activate Stat/JAK. More rigor in pathway analysis is needed to distinguish between these and other possibilities.

It is not clear if the effects observed are due to moving up the lethality curve of Cry9 (i.e. loss of JNK makes insects more sensitive to Cry9), or if these are distinct effects (i.e. JNK is a protective mechanism downstream of Cry9 damage). For example, in Fig 4E, how large are the surviving larvae from a treatment at the LD80 of Cry9A and how much do their intestines thin?

Fig 7 For the insect control assay, the authors sprayed the plants once with a large amount of Bt and dsRNA mix, and then sprayed the insects a second time. This dosing is seems excessive, and it is not clear this would match control in the field. The experiment needs to be repeated with a dose more likely to be used and without spraying the insects themselves. It is concerning that at these doses, only 50-60% mortality was observed in the best case scenario. Also Fig 7E needs the control plants that got no dsRNA with and without insects included and ANOVA for analysis of statistics.

Key issues

Multiple comparisons require ANOVA (with logit or arcsin transformation for percentile data) instead of Student’s t test.

In Figure S1, the authors report >50% mortality at both 5 ug/g and 25 ug/g, yet in Fig 1, the LC50 for the 1st instar larvae is ~50 ug/g. This discrepancy needs to be addressed, and full dose-response curves shown.

Fig 1G, 4D Total Histone H3 is needed as a control for phospho-H3. For Fig 1G, samples should all be re-run on the same gel so they can be compared.

Figs 2, 3, 5, a dose-response curve for Cry9 is needed with the RNAi to see how much the kill curve is shifted by JNK or JAK inhibition or by RNAi.

Fig 5A, a change in mortality from 0.3% to 0.6% is not biologically relevant. Or is this fraction killed, and it is 30% to 60% death?

Reviewer #3: RNA interference of genes that clearly have other functions other than inducing stem cell proliferation raises the problem of separating a specific effect due to Cry toxin intoxication from other general effects. In Figure 7B, there is a difference in mortality even as early as day 1, before the effects of midgut generation could be seen. How are effects of dsStat93E due to other pathways ruled out?

**Part III – Minor Issues: Editorial and Data Presentation Modifications**

Reviewer #1: Minor points:

• To tie the work firmly to the new pesticidal protein nomenclature, it would be useful if, at least once in the manuscript, the full toxin names for the proteins used were listed. Eg is Cry9A, Cry9Aa1 or another variant. Similarly for Vip3A and Cry1F. This detail may be important in the future to allow comparison with the activities of other Cry9A, Cry1F or Vip3A variants. By specifying the toxin variants used in this work, readers will also be able to understand the D125R mutant more clearly if they wish to find the original, wild-type sequence. If the authors do not currently have official BPPRC names for the proteins used, they can be obtained rapidly by submission of sequences through the BPPRC web pages.

• Line 130 (also elsewhere eg lines 140, 397, 405, 411, 717, 718) -please check throughout: gram should be abbreviated to g not gr

• Lines 139 (and throughout): the authors use the terms LC20 and LC2. These are somewhat unusual terms (we usually deal with LC50 or LC90) and it would be useful to define them for the reader on first occurrence in the text. The methods section describes the calculation of LC50 but does not mention calculation of LC20 and LC2 values. I assume that these descriptions LC20/LC2 in the text are to be interpreted as concentration required to give lethal effect in 20%/2% of tested insects. It would be useful if their calculation was described. If this is not what is intended by the terms LC20/LC2, please define them and how they were worked out.

• Lines 154-157: APN has been used previously to measure cell shedding and the method is used here. The description in the result or in the methods does not make clear whether there is a positive or negative correlation of activity with cell shedding. This should be clarified -I assume from the graph that it is a positive correlation (otherwise possible use of APN as a Cry9A receptor and any reduction in activity due to toxin binding may need to be discussed in relation to these results). Fig1F y axis should be labelled as p-nitroanalide released not as cell shedding as p-nitroanalide release is what was actually measured and the units of OD relate to p-nitroanalide absorbance; cell shedding is implied but not measured.

• Fig2B and C: The line for +Cry9A is rather confusing. It appears to be directly beneath bars 1-4 and might be interpreted as referring to those bars. In fact I think it refers to 3-6 and it is approximately under the sloping captions for these bars. Please find a clearer way to annotate

• Line 270: stem striped borer -> striped stem borer

• Fig 6 legend, I would suggest changing “Pore-forming toxins from Bacillus thuringiensis …” to “Pore-forming toxins such as Cry9A, Cry1F and Vip3A from Bacillus thuringiensis …”. The involvement of the proposed pathway has not been demonstrated with Bt toxins such as App or Tpp pore-forming toxins, for example, and it would be best not to overgeneralise beyond what has been shown here.

Reviewer #2: Were dead insects excluded from the analysis of gut thinning? If not, live vs dead insects needs to be separated out.

In the figure legends, it is not clear if 1st or 3rd instar larvae were used for most experiments.

The methods state that 500 ug of RNA was used for cDNA synthesis. This seems high. Is this a typo?

Fig 1A, the full dose-response curves should be shown.

Fig 1C, D, the quantitation does not match the images shown. For example, in 1C the 20 ug dose at 24 h is thinner than the 200 ug dose at 24 h, but 1D shows the 200 ug dose had more thinning.

Fig 1H, how were there 3 biological replicates from 15 larvae? Is the RNA from multiple larvae pooled per experiment? If so, how many are pooled?

Fig 2, the second labels (Cry toxin or time) do not match the conditions because they are shifted too far to the left.

Fig 2E, F, the scale should go to 0 not 40.

Quantitation of Fig 4E and 5H are needed, and RNAi larvae without Cry toxin are needed as controls.

In Fig S2, the Stat92E knockdown is only 50%-70% by RNA. Was there an impact on protein levels? Or could this be an off-target effect?

Reviewer #3: please include the Lepidopteran species names in the keywords

line 73, induces, not induce

twice the following phrase is used, " increased the larvae susceptibility to the treatment with Cry9A ". This is misleading because the susceptibility to Cry9A, an intrinsic property of the species and strain, is not changed. Mortality may have increased or growth may have decreased, but the susceptibility to Cry9A has not changed. "susceptibility to the TREATMENT with Cry9A" adds to the ambiguity. Please choose use a more objective phrase.

PLOS authors have the option to publish the peer review history of their article (what does this mean?). If published, this will include your full peer review and any attached files.

Reviewer #1: **Yes: **Colin Berry

Reviewer #2: No

Reviewer #3: No
---

## [Decision Letter · Decision Letter 1]

13 Nov 2023

Dear Professor Zhang,

We are pleased to inform you that your manuscript 'JAK/STAT signaling regulated intestinal regeneration defends insect pests against pore-forming toxins produced by Bacillus thuringiensis' has been provisionally accepted for publication in PLOS Pathogens.

Best regards,

Raffi V. Aroian

Academic Editor

PLOS Pathogens

Michael Wessels

Section Editor

PLOS Pathogens

Kasturi Haldar

Editor-in-Chief

PLOS Pathogens

orcid.org/0000-0001-5065-158X

Michael Malim

Editor-in-Chief

PLOS Pathogens

orcid.org/0000-0002-7699-2064

Reviewer Comments (if any, and for reference):

Reviewer's Responses to Questions

**Part I - Summary**

Reviewer #1: As described in my first review, the manuscript proposes the reduction of gut defences to Bt toxins will enhance their effects. The involvement of JNK/JAK/STAT in midgut regeneration is established in Chilo suppressalis and JAK/STAT and JNK silencing is shown to potentiate toxin action through a combination of silencing and inhibition approaches.

It is significant in indicating mechanisms by which this insect may recover from or be tolerant to the Cry9 toxin.

Reviewer #3: The authors have addressed my comments on the earlier version to my full satisfaction.

**Part II – Major Issues: Key Experiments Required for Acceptance**

Reviewer #1: The authors appear to have addessed my comments from the first submission along with those raised by other reviewers. In some cases they have indicated that further experiments would be beyond the scope of the current work. I will leave the reviewer concerned to respond to this but I am sympathetic that this is a good and interesting study that I would like to see published and that the full range of experiments may not be feasible in the short term

Reviewer #3: (No Response)

**Part III – Minor Issues: Editorial and Data Presentation Modifications**

Reviewer #1: None

Reviewer #3: (No Response)

PLOS authors have the option to publish the peer review history of their article (what does this mean?). If published, this will include your full peer review and any attached files.

Reviewer #1: No

Reviewer #3: No

---

## [Editor Report · Acceptance letter]

4 Dec 2023

Dear Professor Zhang,

We are delighted to inform you that your manuscript, "JAK/STAT signaling regulated intestinal regeneration defends insect pests against pore-forming toxins produced by Bacillus thuringiensis," has been formally accepted for publication in PLOS Pathogens.

Best regards,

Michael Malim

Editor-in-Chief

PLOS Pathogens

orcid.org/0000-0002-7699-2064